# The Fusion Gene Landscape in Taiwanese Patients with Non-Small Cell Lung Cancer

**DOI:** 10.3390/cancers13061343

**Published:** 2021-03-16

**Authors:** Ya-Sian Chang, Siang-Jyun Tu, Ju-Chen Yen, Ya-Ting Lee, Hsin-Yuan Fang, Jan-Gowth Chang

**Affiliations:** 1Epigenome Research Center, China Medical University Hospital, Taichung 404332, Taiwan; t25074@mail.cmuh.org.tw (Y.-S.C.); t24399@mail.cmuh.org.tw (J.-C.Y.); t23701@mail.cmuh.org.tw (Y.-T.L.); 2Department of Laboratory Medicine, China Medical University Hospital, Taichung 404332, Taiwan; t34752@mail.cmuh.org.tw; 3Center for Precision Medicine, China Medical University Hospital, Taichung 404332, Taiwan; 4Department of Medical Laboratory Science and Biotechnology, China Medical University, Taichung 404333, Taiwan; 5Department of Thoracic Surgery, China Medical University Hospital, Taichung 404332, Taiwan; d17573@mail.cmuh.org.tw; 6School of Medicine, China Medical University, Taichung 404333, Taiwan; 7Department of Bioinformatics and Medical Engineering, Asia University, Taichung 41354, Taiwan

**Keywords:** NSCLC, RNA-sequencing, fusion genes, kinase, druggable

## Abstract

**Simple Summary:**

Human cancer genomes show a variety of alterations, such as single base changes, deletions, insertions, copy number changes, and gene fusions. Analyzing fusion gene transcripts may yield a novel and effective approach for selecting cancer treatments. However, few comprehensive analyses of gene fusions in non-small cell lung cancer (NSCLC) patients have been performed. Here, we characterized the fusion gene landscape of NSCLC in a case study of Taiwanese lung cancer patients. We concluded that some fusion genes likely play driver roles in carcinogenesis, while others act as passengers. We demonstrated that by using RNA-sequencing to detect gene fusion events, putative therapeutic drug targets could be identified, potentially leading to more precise therapies for NSCLC.

**Abstract:**

Background: Analyzing fusion gene transcripts may yield an effective approach for selecting cancer treatments. However, few comprehensive analyses of fusions in non-small cell lung cancer (NSCLC) patients have been performed. Methods: We enrolled 54 patients with NSCLC, and performed RNA-sequencing (RNA-Seq). STAR (Spliced Transcripts Alignment to a Reference)-Fusion was used to identify fusions. Results: Of the 218 fusions discovered, 24 had been reported and the rest were novel. Three fusions had the highest occurrence rates. After integrating our gene expression and fusion data, we found that samples harboring fusions containing *ASXL1*, *CACNA1A*, *EEF1A1*, and *RET* also exhibited increased expression of these genes. We then searched for mutations and fusions in cancer driver genes in each sample and found that nine patients carried both mutations and fusions in cancer driver genes. Furthermore, we found a trend for mutual exclusivity between gene fusions and mutations in the same gene, with the exception of *DMD*, and we found that *EGFR* mutations are associated with the number of fusion genes. Finally, we identified kinase gene fusions, and potentially druggable fusions, which may play roles in lung cancer therapy. Conclusion: The clinical use of RNA-Seq for detecting driver fusion genes may play an important role in the treatment of lung cancer.

## 1. Introduction

Lung cancer is the most common malignant neoplasm in humans and the leading cause of cancer mortality worldwide [1]. Tobacco exposure is the number one risk factor for lung cancer [2]. Lung cancer is classified into different histological subtypes: non-small cell lung cancer (NSCLC) and small cell lung cancer. NSCLC accounts for 85% of all lung cancer diagnoses [3,4]. NSCLC has different characteristics in different ethnic groups; for example, the mutation frequencies of *EGFR*, *KRAS*, *BRAF*, and *TP53* differ significantly between Asian and Western populations [5], which may result in different therapeutic outcomes.

Promising molecular-targeted therapies have been developed for the treatment of lung cancer. Therapies that are currently on the market include epidermal growth factor receptor (EGFR), tyrosine kinase inhibitors (gefitinib (Iressa^®^), erlotinib (Tarceva^®^), afatinib (Gilotrif^®^), and osimertinib (Tagrisso^®^)), anaplastic lymphoma kinase (ALK) inhibitors (crizotinib (Xalkori^®^), ceritinib (Zykadia^®^), and alectinib (Alecensa^®^)), vascular endothelial growth factor-targeted monoclonal antibodies (bevacizumab (Avastin^®^) and ramucirumab (Cyramza^®^)), and anti-PD-1/anti-PD-L1 antibodies (pembrolizumab (Keytruda^®^), nivolumab (Opdivo^®^), atezolizumab (Tecentriq^®^), and durvalumab (Imfinzi^®^)) [6]. However, resistance to cancer therapies remains a major clinical problem and involves all models of treatment, including molecular-targeted therapy and immunotherapy.

Detecting changes in human cancer genomes is crucial for developing targeted therapies for precision medicine. Most recent studies have focused on detecting mutations, insertions, deletions, and copy number changes when examining cancer genomes; broadening the search to include fusion transcripts may better aid cancer treatment selection. The prevalence of gene fusions in human cancers is approximately 20% [7]. However, the frequency of gene fusions varies greatly among cancer types, and most gene fusions are tumor specific, which supports the need for different oncogenic and therapeutic approaches. Examples of common gene fusions in cancer include *EWSR1*-*FLI1* in Ewing’s sarcoma [8], *BCR*-*ABL1* in chronic myeloid leukemia [9], and *DNAJB1*-*PRKACA* in fibrolamellar carcinoma [10]. Of the 28 Food and Drug Administration-approved drugs, 11 were approved to target fusion genes (*ALK*, *ROS1*, *NTRK*, *PML*-*RARA* and *BCR*-*ABL*), 16 were approved to target non-fusion genes (*EGFR*, *BRAF*, *BRCA*, *HER2*, *FLT3*, *IDH1*, and *IDH2*), and one was approved to target both fusion and non-fusion genes (the fusion target *BCR*-*ABL* and the non-fusion target *KIT*) [11]. Fusion genes are therefore promising targets for the treatment of cancer. 

The accurate detection of gene fusions in patients has the potential to improve patient care. Multiple methods have been developed to detect gene fusions, such as fluorescence in situ hybridization, quantitative real-time polymerase chain reaction, G-banding cytogenetics, and immunohistochemistry. These methods are time-consuming and labor intensive and can detect only one or several known fusion transcripts. Advancements in next-generation sequencing technology have enabled the rapid assessment of many potential gene fusions in a single test, including known and uncharacterized gene fusions via RNA-Sequencing (RNA-Seq), which is currently the most common method for analyzing multiple fusion genes on a genomic scale. 

Previous studies typically used The Cancer Genome Atlas (TCGA) RNA-Seq database to analyze fusions in different tumor types [12,13,14], rarely using their own data. Rudin et al. [15] performed the first genome-wide comprehensive analysis of gene fusions in lung cancer patients using their own case study. They identified multiple fusion transcripts and a recurrent fusion of *RLF*-*MYCL1*. Subsequently, Sao et al. found 45 fusion genes in 87 lung cancer patients, 8 of which were chimeric tyrosine kinase genes (*EML4*-*ALK*, *KIF5B*-*RET*, *CD74*-*ROS1*, *SLC34A2*-*ROS1*, *CCDC6*-*ROS1*, *SCAF11*-*PDGFRA*, *FGFR2*-*CIT*, and *AXL*-*MBIP*), which play important roles in cancer development [16]. In another report, the gene fusions of six never-smoking female patients with lung cancer were analyzed by RNA-Seq; these patients had no mutations in *KRAS* or *EGFR* and were negative for *EML4*-*ALK* [17]. Recently, Miyanaga et al. identified eight fusion genes in six pulmonary carcinoid patients using RNA-Seq [18].

In this study, we performed RNA-Seq to identify fusion events in a population of Taiwanese patients with NSCLC. Moreover, we investigated the relationship between the fusion status and gene expression/mutation status of genes.

## 2. Results

### 2.1. Identification of Fusion Genes in NSCLC

We performed RNA-Seq in samples resected from 54 NSCLC patients (54 tumor tissues and 8 matched normal tissues) and applied a screening process to identify fusion genes in NSCLC (Figure 1). Using STAR-Fusion, we identified 311 fusion genes. Next, we filtered reads using additional criteria: excluding fusions that were expressed in normal tissues, number of reads > 10, and reads with an intrachromosomal rearrangement distance cutoff > 10 kb. The fusion types included in the search were fusions between two protein coding genes, between two long non-coding RNAs (lncRNAs), of a protein coding gene followed by an lncRNA, and of an lncRNA followed by a protein coding gene. This study identified 218 fusion genes from 49 samples (Appendix A); 24 of these fusions have been reported: *AC010998.1*-*WDR11-AS1*, *ACOT9*-*APOO*, *AHNAK*-*NXF1*, *AL135923.2*-*DMAC1*, *AL445647.1*-*MIR4500HG*, *CASC19*-*CASC8*, *FANCA*-*FP236383.1*, *FGFR3*-*TACC3*, *FRAS1*-*MRPL1*, *FTH1*-*FTL*, *JMJD1C-AS1*-*REEP3*, *KIF5B*-*RET*, *KRT5*-*KRT17*, *LINC02159*-*ATP10B*, *MTMR3*-*UQCR10*, *MUC4*-*TNK2*, *PSAP*-*SFTPC*, *RAB31*-*PPP4R1-AS1*, *SBNO1*-*RILPL1*, *SLC12A2*-*CCDC192*, *TBC1D15*-*RAB21*, *TMEM123*-*YAP1*, *TPH2*-*TRHDE*, and *UBAP2*-*DCAF12*. These 24 known gene fusions were present in 17 of our NSCLC patients (Appendix A). We validated 77 novel fusion genes using reverse transcription-polymerase chain reaction (RT-PCR) and Sanger sequencing (Appendix A).

To identify novel fusions that paly roles in cancer development, we selected transcripts predicted to encode protein products (108 fusion genes) and then applied the IUPred2A tool to predict protein functional changes. As a result, 25 fusion proteins were predicted to exhibit disordered functions (Appendix A) and were found in 15 of the NSCLC patients. We also calculated the number of important mutants in 299 known cancer driver genes (CDGs) [19] in each case. The average number of important variants in each case was approximately three. Nine patients had more than the average number of CDG variants, suggesting that the fusion genes in these patients may play a passenger role as traditional passenger variants. However, six patients, in whom seven novel gene fusions (*NUP153*-*DHX16*, *PFDN4*-*ASXL1*, *PPP2R5A*-*NENF*, *RBM19*-*BRAP*, *HTT*-*CDH12*, *SFTPA1*-*FTL*, and *PPP1R37*-*THRA*) were found, had fewer than average important CDG variants, suggesting that these seven fusion genes may play a driver role as traditional driver variants.

Three fusion genes exhibited particularly high occurrence rates (=number of patients with fusion genes/49): *FTH1*-*FTL*, *ARHGAP21*-*PRTFDC1*, and *LINC02159*-*ATP10B*. The occurrence rate was approximately 6.12% (3/49) for *FTH1*-*FTL* and 4.08% (2/49) for the other two fusions. Other fusion genes were mutually exclusive. The *FTH1*-*FTL* and *LINC02159*-*ATP10B* fusions have previously been reported in the Cancer Cell Line Encyclopedia and TCGA datasets; however, their functions have not been well characterized.

### 2.2. The Clinical Significance of Fusion Genes Involving Oncogenes or TSGs

Next, we grouped the fusion genes into those involving oncogenes or TSGs and then determined which, if any, of the fusion genes were expression outliers. Five of the fusion genes involved a TSG (*PFDN4*-*ASXL1*, *BCOR*-*DMD*, *CDK12*-*PNMT*, *CSDE1*-*TERT*, and *KANSL1*-*AC091132.4*) and were found in five patients. The TSG *ASXL1* was the only observed expression outlier, i.e., *ASXL1* was aberrantly expressed in the samples harboring *ASXL1* fusion genes (Figure 2A and Appendix A). In five patients, seven fusion genes involved an oncogene (*FGFR3*-*TACC3*, *EEF1A1*-*RPL3*, *DMD*-*PHEX*, *ACOT9*-*DMD*, *BCOR*-*DMD*, *DHX34*-*CACNA1A*, and *KIF5B*-*RET*). Three of these oncogenes (*CACNA1A*, *EEF1A1*, and *RET*) were classified as expression outliers (Figure 2B–D and Appendix A).

The clinicopathological features of the NSCLC patients with fusion genes involving TSGs (tumor suppressor genes) and/or oncogenes were investigated. We found that age and grade were correlated with the presence of fusion genes involving TSGs (*p* = 0.0208 and *p* = 0.0363, respectively), whereas grade was correlated with the presence of fusion genes involving oncogenes (*p* = 0.0086) (Table 1). Patients with a higher grade have a higher relative percentage of fusion genes involving oncogenes and TSGs, but no correlation with tumor stage. No significant correlation was found between overall survival and the presence of fusion genes involving TSGs or oncogenes (Appendix AA,B).

### 2.3. Mutual Exclusivity of Gene Fusions and Gene Mutations

Mutations in oncogenes and TSGs play key roles in cancer induction, and fusions involving such genes are likely to contribute to cancer development. To better understand how fusion genes contribute to carcinogenesis in NSCLC patients, we systematically profiled mutations and fusions in 299 known CDGs (cancer driver genes) [19]. We then classified the patients into three groups (some overlapping): those with CDG mutations known to be pathogenic or likely to be pathogenic (according to the ClinVar database), those with CDG mutations with predicted pathogenicity, and those with a gene fusion involving a known CDG.

Our results showed that 77.78% (42/54) and 22.22% (12/54) of the patients carried known driver mutations and predicted pathogenic driver mutations, respectively (Appendix A). Of the 42 patients with known driver mutations, seven carried both driver mutations and driver fusion events. Of the 12 patients with predicted pathogenic mutations only, two had both a fusion gene and a predicted pathogenic mutation (Figure 3).

Of the 35 patients with known driver mutations only, the total number of variants per sample ranged from 1 to 8, averaging 3.91. The total number of variants per sample in the seven patients with known driver mutations and fusions ranged from 1 to 5, with an average of 3.00. The samples with fusions carried fewer mutations compared with those without fusions, although this difference was not significant. In the 10 patients with predicted pathogenic mutations only, the total number of variants per sample ranged from 1 to 5, with an average of 2.8, which was lower than the average number recorded in the patients with known driver mutations. There was only one variant per sample in the two patients with predicted pathogenic mutations and fusions. Similar to the patients with known driver mutations, samples harboring fusions of predicted CDGs had fewer mutation variants.

We then examined copy number variations in the 10 patients who carried predicted pathogenic mutations only. Seven of these patients had increased copy numbers of oncogenes and decreased copy numbers of TSGs (Table 2). These data suggest that the predicted pathogenic variants in the remaining three patients likely act as true driver mutations.

We further examined the relationship between fusions and mutations in the same driver gene. Of the samples from eight patients, 10 harbored fusion genes that differed from the genes with mutations. Only one patient carried a fusion and mutation involving the same gene (*DMD*). This suggests that the majority of gene fusions and mutations are mutually exclusive and do not tend to occur in the same gene.

### 2.4. Correlation between EGFR Mutations and the Number of Fusion Genes 

*EGFR* mutations play important roles in the development and treatment of lung cancer, and they occur frequently in Taiwanese people and other Asians. The relationship between *EGFR* mutations and fusion genes has not been explored previously. We investigated whether fusion genes are correlated with *EGFR* mutations. The results showed that the *EGFR* mutations are not associated with driver fusions (Table 3). However, we found that *EGFR* mutations are associated with the number of fusion genes (*p* = 0.0038), and patients with *EGFR* mutations may have more fusion genes (≥5) than patients with wildtype *EGFR*. 

### 2.5. Structure and Spectrum of Kinase Gene Fusions

In total, we detected 21 kinase gene fusions, of which 12 contained the kinase at the 5′ end (*CAMK1D*, *CDK12*, *DGKZ*, *DMPK*, *DYRK4*, *FGFR3*, *MAGI1*, *MAST4*, *PTK2*, *RPS6KA1*, *SMG1*, and *TAOK1*), and nine at the 3′ end (*ERN2*, *GUCY2C, MAGI1*, *NIM1K*, *RET*, *STK10*, *TEK*, *TNK2*, and *WNK3*). Analysis of the catalytic kinase domains using the PFAM domain database showed that six of the kinase gene fusions retained an intact kinase domain (*CDK12*, *DMPK*, *ERN2*, *FGFR3*, *NIM1K*, and SMG1). We then used AGFusion software to check whether the annotated kinase domain is still present in the fusion transcripts, allowing us to classify the fusions as those with an intact kinase domain versus a disrupted kinase domain. Three fusions (*DMPK*-*SYMPK*, *FGFR3*-*TACC3*, and *SMG1-ERN2*) were in-frame, whereas *CARMIL1*-*NIM1K* and *CDK12*-*PNMT* were out-of-frame. Four of the fusions (*CARMIL1*-*NIM1K*, *CDK12*-*PNMT*, *FGFR3*-*TACC3*, and *SMG1-ERN2*) had intact kinase domains. The breakpoint position of five fusions was within the coding sequences (Figure 4).

### 2.6. Contributions of Fusions to Cancer Treatment

Using the Database of Evidence for Precision Oncology, we identified clinically relevant genomic fusion alterations in five patients (10.20%). Clinically relevant fusion alterations were found in the following genes: *FGFR3* (1, 1.85%), *NOTCH2* (2, 3.70%), *TMPRSS2* (1, 1.85%), and *RET* (1, 1.85%). The drugs for *FGFR3*, *NOTCH2*, and *TMPRSS2* were off-label use, while *RET* was on-label use. 

We also examined drug-related fusion genes using our RNA-Seq data, including *ALK*, *ROS1*, *NTRK1/2*, *EGFR*, *BRAF*, and *RET*, and only one *RET* fusion was found, which might have been due to the small sample size.

## 3. Discussion

Our study explored the clinical significance of fusion genes in lung cancer for the first time in Taiwan. We examined whether the concepts of driver and passenger fusions are useful in explaining the importance of fusion genes in the development of lung cancer, just as driver and passenger mutations. Therefore, we classified novel fusions as driver or passenger fusions according to the number of detected cancer driver gene variants. If a patient has fewer than the average number of gene variants and no known driver variants, any fusion gene observed is called a driver fusion. By contrast, if the patient has more than the average number of gene variants, especially known driver variants, any fusion gene observed is called a passenger fusion.

We found that 16.66% of the patients (9/54) harbored fusions involving CDGs, together with known cancer driver mutations (seven patients) or predicted pathogenic variants (two patients). We then examined the combined oncogene mutations in nine samples and found mutations in *PIK3CA*, *FGFR2*, *EGFR*, *DMD*, *PTPDC1*, and *DIAPH2* in patients who also harbored driver fusions. We therefore speculated that these genes are relatively weak oncogenes that require additional alterations for progression to carcinogenesis (Appendix A). Moreover, no gene fusions involving CDGs were found in patients with mutations in the oncogenes *CTNNB1*, *DACH1*, *ERBB2*, *FGFR3*, *GNAS*, *GTF2I*, *KLF5*, *KRAS*, *MSH3*, and *POLRMT*. This suggests that these genes are relatively strong oncogenes that drive cancer development without the need for additional driver gene fusions (Appendix A). Choi et al. reported the presence of somatic mutations in various TSGs in colorectal cancer patients who also carried fusion genes, whereas no non-synonymous somatic oncogene mutations were observed in patients harboring fusion genes [20]. In this study, we found that somatic mutations in various oncogenes and TSGs were present in NSCLC patients positive for fusion genes, although our results differed from those of Choi et al.; we propose that the strength of the oncogenic effects of genes should be considered. Some weak oncogenic variants require other oncogenes, TSGs, or fusion genes to drive cancer. In addition, Choi et al. did not find any somatic mutations within either the partner gene of a fusion [20]. Similarly, our results suggest that mutations are rarely present in fusion genes. Only one patient with a mutation in the *DMD* gene had a concomitant fusion involving *DMD*; this implies that the oncogenic effect of the *DMD* mutation or fusion is weak when present alone.

The presence of fusion genes may result in altered expression of one or both of the fusion gene partners [21]. Samples harboring fusions containing TSGs (*ASXL1*) or oncogenes (*CACNA1A*, *EEF1A1*, and *RET*) exhibited increased expression of those genes relative to the samples without fusions. Gao et al. demonstrated that fusion genes influence oncogene expression, usually via overexpression of oncogenes and reduced expression of TSGs [7]. Further investigation is needed to determine whether the dysregulated oncogene expression is caused by the fusion events or another factor, as in the study by Lu et al. [22].

In hematopoietic malignancies and solid tumors, gene fusions can result in kinase activation. Kinases represent a major class of oncogenes, and two well-known examples of cancer-promoting kinase gene fusions are the *BCR*-*ABL1* fusion in leukemia patients and fusion of *ALK* to various other proteins in lung carcinomas and mesenchymal tumors. Drugs that target these two kinases are now available [23,24]. In this study, we found 21 kinase gene fusions, which we translated into peptide sequences; five of these kinase gene fusions had intact functional catalytic domains (*DMPK*-*SYMPK*, *FGFR3*-*TACC3*, *CARMIL1*-*NIM1K*, *CDK12*-*PNMT*, and *SMG1*-*ERN2*). Our findings could provide a useful reference for cancer drug development. *FGFR3*-*TACC3* is a common fusion in several cancer types [14]. Therefore, we propose *FGFR3*-*TACC3* as a candidate target for drug development to treat NSCLC.

A correlation was observed for the grade and fusion genes involving TSGs and oncogenes but no correlation with tumor stage. Fusion genes involving TSGs and oncogenes in the tumors at higher grades were significantly more frequent than those in the tumors at lower grades. These results have also been found in prostate and brain tumors in the specific fusion genes *TMPRSS2*-*ERG* and *ARHGEF2*-*NTRK1*, which involved *ERG* and *NTRK1* oncogenes [25,26]. We also explored the correlation between *EGFR* mutations and the number of fusion genes in each case. The results showed that cases with *EGFR* mutations had more fusion genes. The association between the occurrence of fusion genes and the *EGFR* genetic background may result from tumors with an *EGFR* mutation exhibiting increased cell proliferation and survival, which then increases the likelihood of fusion genes occurring [27].

This study has the following limitations. First, the cohort size was too small to make more definitive conclusions and obtain statistical significance in the correlation analysis; several gene fusions were found in three or fewer patients, making population-wise interpretation almost impossible. Second, we did not perform functional studies to confirm the effects of driver fusion genes, which might have influenced our subgrouping accuracy. Third, the study is based solely on the Taiwanese population. The results should be re-evaluated in other studies based on larger heterogeneous cohorts, to further validate our findings.

## 4. Materials and Methods 

### 4.1. Patient Samples

Tissue specimens were obtained from 54 Taiwanese patients with lung cancer who had undergone surgical resection between May 2007 and April 2014 at the China Medical University Hospital. Of these patients, eight had matched tumor and normal tissues. The 54 lung tumors comprised 44 adenocarcinomas and 10 squamous cell carcinomas. The surgically resected specimens were immediately dissected and preserved in liquid nitrogen. The present study was approved by the Institutional Review Board of the China Medical University Hospital (CMUH106-REC1-053).

### 4.2. RNA-Seq

Total RNA was extracted from the clinical tissue samples using the NucleoSpin^®^ RNA Kit (Macherey-Nagel, Düren, Germany) according to the manufacturer’s instructions. The quality, quantity, and integrity of the total RNA were evaluated using a NanoDrop1000 spectrophotometer and Agilent 2100 Bioanalyzer (Agilent Technologies, Santa Clara, CA, USA). Samples with an RNA integrity number > 6.0 were used for RNA-Seq. A barcoded mRNA library was generated using the TruSeq Stranded mRNA Library Preparation Kit (Illumina, San Diego, CA, USA). The libraries were sequenced on the Illumina Nova Seq 6000 (Illumina) using 2 × 151 bp paired-end sequencing flow cells, according to the manufacturer’s instructions.

### 4.3. RNA-Seq Data Analysis

We used Trimmomatic for quality control of the RNA-Seq data [28], as described previously [29]. For gene expression, the reads that passed quality control were aligned with the GRCh38 human genome using HISAT2 [30], and featureCounts was used to quantify gene expression with annotation done using GENCODE v22 without mitochondrial genes [31]. The transcripts-per-million normalization method was applied. 

To detect fusion transcripts, STAR-Fusion was applied using reads that passed quality control and the CTAT library downloaded from https://data.broadinstitute.org/Trinity/CTAT_RESOURCE_LIB/__genome_libs_StarFv1.7/ (accessed on 17 August 2019). FusionInspector, a sub-module of STAR-Fusion, was used for IGV validation and annotation [32]. To remove false-positive fusions, fusions reported in eight tumor-adjacent normal tissues were filtered out. Furthermore, only fusion read counts > 10 were included; the fusion read count was calculated by summing the “JunctionReadCount” and “SpanningFragCount” from the STAR-Fusion report. If a fusion rearrangement was present in the same chromosome, the distance between the left and right breakpoints had to be > 10 kb. ChimerDB2, ChimerKB, ChimerPub, ChimerSeq, Cosmic, YOSHIHARA TCGA [33], Klijin CellLines [34], GUO2018CR TCGA [7], TumorFusionNAR2018 [13], TCGA StarF2019 (fusions found by STAR-Fusion v1.5.0 across corresponding TCGA datasets), and CCLE StarF2019 (fusions identified in the Cancer Cell Line Encyclopedia RNA-Seq datasets) were used to find reported gene fusions. Novel fusion protein disorders were detected using IUPred2A [35]. We used the PFAM domain database (https://pfam.xfam.org/, accessed on 2 November 2020) to check for genes annotated with a kinase domain and AGFusion to examine fusion transcripts with an intact or disrupted kinase domain [36]. Fusion events and their respective drug therapies were found using the Database of Evidence for Precision Oncology [37].

### 4.4. Reverse Transcription-Polymerase Chain Reaction and Sanger Sequencing

Fusion genes were validated using RT-PCR with primers that target fusion gene breakpoints. Total RNA was extracted from the clinical tissue samples using the NucleoSpin^®^ RNA Kit (Macherey-Nagel). RNA samples (2000 ng) were reverse-transcribed into cDNA using High-Capacity cDNA Reverse Transcription kits (Applied Biosystems, Foster City, CA, USA). The PCR primers are shown in Appendix A. The PCR program consisted of 5 min at 95 °C, 35 cycles of 30 s at 95 °C, 30 s at 52–60 °C, and 30 s at 72 °C, and a final 7 min at 72 °C. All the Sanger sequencing experiments were performed at Genomics (https://www.genomics.com.tw/, accessed on 24 December 2020).

### 4.5. Whole Exome Sequencing and Data Analysis

A total of 50 ng of DNA (based on Qubit quantification) were tagged and fragmented by a transposome and then cleaned up and amplified. A 200-400-bp band was selected, and exome capture was performed using the Nextera Exome Library Preparation Kit (Illumina, San Diego, CA, USA). The DNA library was quantified using a Qubit 3.0 Fluorometer (Life Technologies (Thermo Fisher Scientific), Waltham, MA, USA) and an Agilent 4200 Bioanalyzer (Agilent Technologies, Palo Alto, CA, USA). Samples were subjected to paired-end sequencing using the Illumina NovaSeq 6000 platform with a 150-bp read length. Details of the whole exome sequencing analysis have been described previously [38].

### 4.6. Copy Number Variation Analysis

DNA copy number profiling was performed using the customized Axiom-Taiwan Biobank Array (TWB chip: Affymetrix, Santa Clara, CA, USA). This customized array included 653,291 single nucleotide polymorphisms that were selected based specifically on the Taiwanese population, and genotyping was performed using standardized procedures of the National Center of Genome Medicine (Academia Sinica, Taipei, Taiwan). Axiom Analysis Suite (http://www.affymetrix.com accessed on 7 January 2020) was used to call the copy number variation in each sample. The copy number variation was annotated using the AnnotSV tool [39].

### 4.7. Statistical Analyses

Clinicopathological samples were analyzed for the presence of fusion genes using Fisher’s exact test. The Kaplan–Meier method was used to construct overall survival curves, which were then compared using the log-rank test. We compared variant numbers between patients with and without gene fusions using Welch’s t-test. *p*-values < 0.05 were considered to indicate significance. Statistical analyses were performed using SPSS v22.

## 5. Conclusions

In this study, we identified known and novel fusion genes present in NSCLC patients using RNA-Seq and integrated gene expression, mutation, and fusion annotation data. Expression levels increased with *ASXL1* (tumor suppressor gene), *CACNA1A*, *EEF1A1*, and *RET* (three oncogenes) fusions. The tumor grade and age were associated with fusion genes containing TSGs and tumor grade was also associated with fusion genes containing oncogenes. Nine patients carried both mutations and fusions of CDGs. Mutual exclusivity between the two types of genomic alteration, fusions and mutations, was observed. Furthermore, our study revealed an association between *EGFR* mutations and the number of fusion genes in the same patient. Some cancer patients may benefit from existing drugs that target fusion protein partners, expanding the therapeutic options available. Routine searches for fusion genes in cancer patients may improve their survival by revealing additional treatment targets for precise personalized medicine.

## Figures and Tables

**Figure 1 cancers-13-01343-f001:**
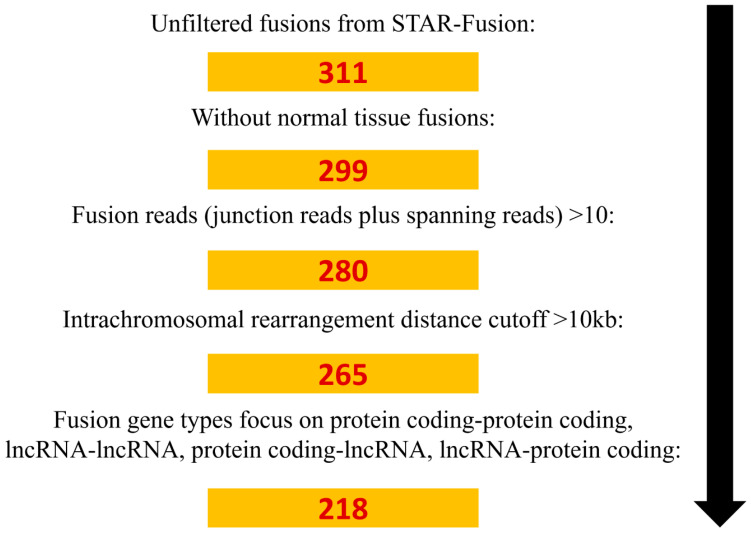
Overview of the filtering process of fusion gene RNA-Seq reads obtained from NSCLC (non-small cell lung cancer) samples.

**Figure 2 cancers-13-01343-f002:**
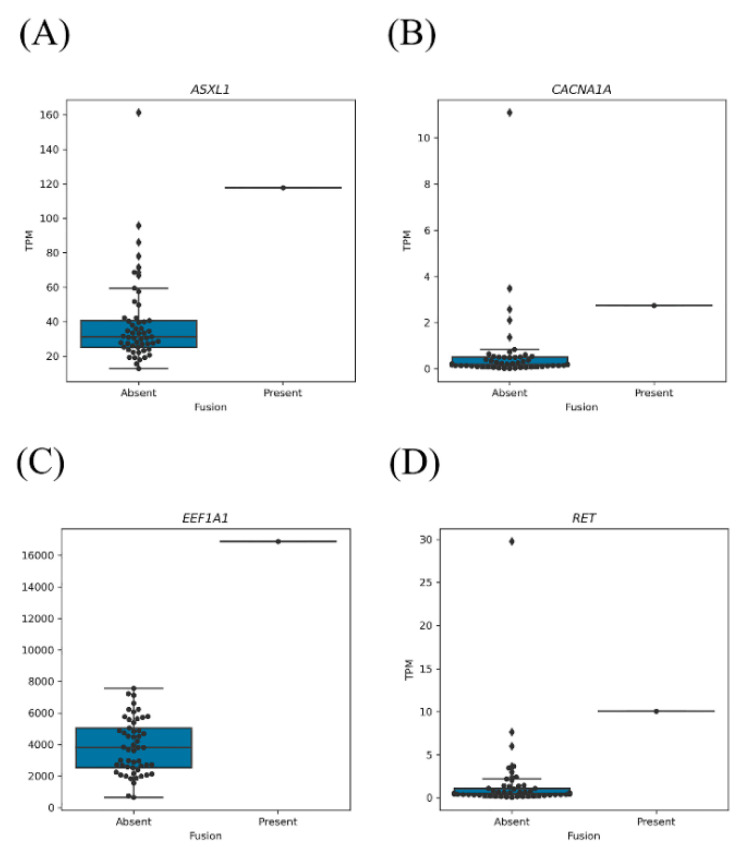
Expression levels of genes present in fusions; expression outliers are shown. (**A**) *ASXL1* expression in samples with or without *ASXL1* fusion genes. (**B**) *CACNA1A* expression in samples with or without *CACNA1A* fusion genes. (**C**) *EEF1A1* expression in samples with or without *EEF1A1* fusion genes. (**D**) *RET* expression in samples with or without *RET* fusion genes.

**Figure 3 cancers-13-01343-f003:**
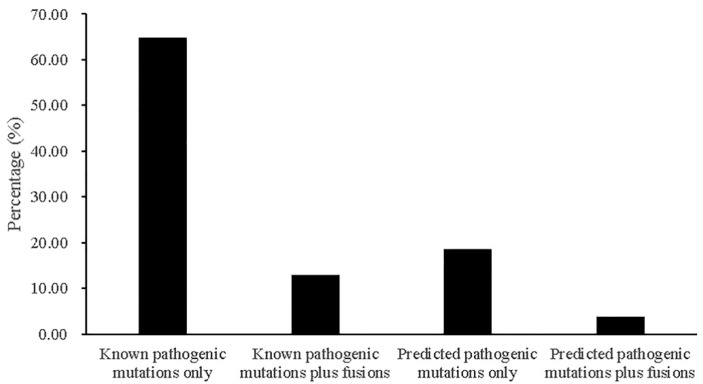
The chart shows the percentages of samples containing known pathogenic mutations only, predicted pathogenic mutations only, known pathogenic mutations plus fusions, or predicted pathogenic mutations plus fusions in 299 known cancer driver genes.

**Figure 4 cancers-13-01343-f004:**
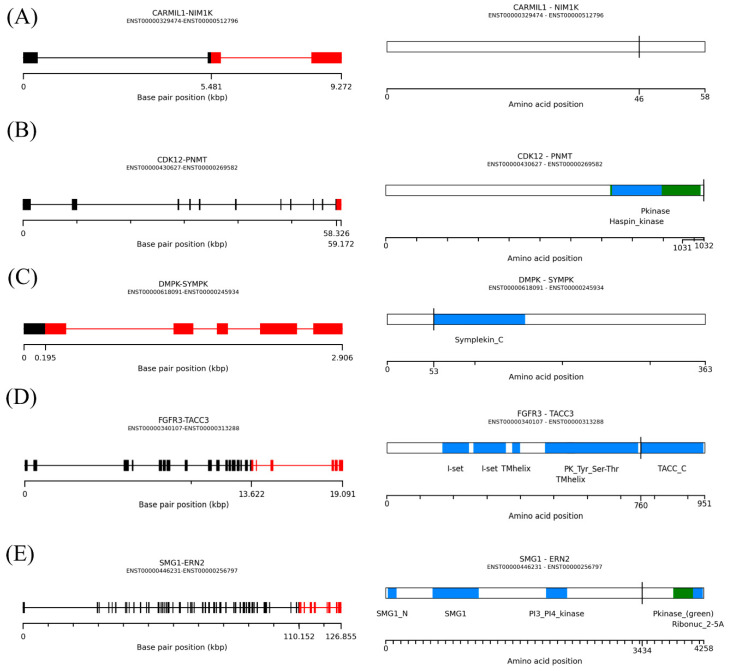
Predicted structures of kinase gene fusions. (**A**) *CARMIL1*-*NIM1K*, (**B**) *CDK12*-*PNMT*, (**C**) *DMPK*-*SYMPK*, *(***D***) FGFR3*-*TACC3*, and (**E**) *SMG1*-*ERN2*.

**Table 1 cancers-13-01343-t001:** Relationships between fusion genes detected using RNA-Seq and clinicopathological characteristics in 54 patients with NSCLC.

Features		Fusion Genes Involving Oncogenes			*p*-Value	Fusion Genes Involving TSGs			*p*-Value
		No	Yes	Total		No	Yes	Total	
Gender	Male	26	2	28	0.6633	26	2	28	0.6633
	Female	23	3	26		23	3	26	
Age	<70	25	3	28	1.000	28	0	28	0.0208
	≥70	24	2	26		21	5	26	
Stage	I-II	34	2	36	0.1626	33	3	36	0.6374
	III-IV	13	3	16		14	2	16	
Grade	1-2	38	1	39	0.0086	38	1	39	0.0363
	3-4	8	4	12		9	3	12	
Histology type	ADC	41	3	44	0.2273	39	5	44	0.5707
	SqCC	8	2	10		10	0	10	
Smoking status	Non-smokers	32	3	35	1.000	31	4	35	0.6461
	Smokers	17	2	19		18	1	17	

*p*-value by Fisher’s exact test. All subjects underwent surgical resection. Patients were diagnosed according to the 2015 World Health Organization and International Association for the Study of Lung Cancer guidelines. Staging was based on the guidelines of the eighth edition of the TNM classification for NSCLC.

**Table 2 cancers-13-01343-t002:** Summary of gains of oncogenes and losses of TSGs (tumor suppressor genes) in seven NSCLC patients.

Sample	Gains of Oncogenes	Losses of TSGs
Lung-T151	*KIT*, *CEBPA*, *PLCB4*	*ATM*, *KMT2A*, *MEN1*, *WT1*, *STK11*
Lung-T218	*DICER1*, *GNAS*, *DMD*	none
Lung-T227	*SMARCA4*	none
Lung-T236	*KIT*, *RET*, *DICER1*, *GNAS*, *COL5A1*	none
Lung-T291	*KIT*, *CARD11*, *WT1*, *DICER1*, *MAX*, *CDH1*, *EP300*	none
Lung-T95	*MSH3*, *PTPN11*, *DICER1*, *CDH1*, *EP300*, *MYH9*	*BAP1*, *ATM*, *CHD8*, *LZTR1*, *CDKN2A*
Lung-T430	*GNAS*, *MACF1*	none

**Table 3 cancers-13-01343-t003:** Correlation between fusions and *EGFR* mutations.

Features	Genotype	Fusion Genes Involving 299 CDGs			*p*-Value	Number of Fusion Genes			*p*-Value
		**No**	**Yes**	Total		<5	≥5	Total	
*EGFR*	Mutation	22	5	27	1.000	12	15	27	0.0038
	Wildtype	23	4	27		23	4	27	

*p*-value by Fisher’s exact test

## Data Availability

The RNA-Seq and WES data from this study were submitted to the NCBI Sequence Read Archive (SRA) under BioProject accession nos. PRJNA698419 and PRJNA698277, respectively.

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
