# Peer review of "The Fusion Gene Landscape in Taiwanese Patients with Non-Small Cell Lung Cancer"

_cancers, 2021, doi:10.3390/cancers13061343_

Round 1
Reviewer 1 Report
The authors substantially improved the manuscript by incorporating the reviewers´ comments and modificated the original text. It is a novel original study with practical output. It is overally well written.
The results are nicely presented with many figures and tables that make the reading easier. The results are adequately discussed and the upto date literature is discussed in the light of the study results and conclusions.
Author Response
Reviewer #1:
The results are nicely presented with many figures and tables that make the reading easier. The results are adequately discussed and the up to date literature is discussed in the light of the study results and conclusions.
Response: Thank you for your comment.
Reviewer 2 Report
I thank the authors for their response to my review. I believe that the authors have addressed my comments satisfactorily. However, I think Figure 2 should be Supplementary Figure.
Author Response
Reviewer #2:
I thank the authors for their response to my review. I believe that the authors have addressed my comments satisfactorily. However, I think Figure 2 should be Supplementary Figure.
Response: Thank you for your comment. Figure 2 has been removed to Figure S1. We have rearranged the order of the Figures and Supplementary Figures.
Reviewer 3 Report
In a revised version of the manuscript Chang et al addressed some of my concerns, however, several suggestions are not addressed, which decreases the impact of the study; authors list them as limitations in the end of the manuscript.
Several additional suggestions:
- After looking at updated Table 1 it becomes prominent, that fusion landscape is quite different in adenocarcinoma and squamous cell carcinoma samples, potentially reflecting differences in biology in between these non-small cell lung cancer subtypes. Additionally, samples with higher stage and grade demonstrate higher relative percentage of fusion genes involving oncogenes and tumor suppressor genes. Authors should address these points in results and discussion sections.
- Figure 2 occupies two pages. It seems to be beneficial to convert it into a table and move seq chromatograms to supplemental figures.
Author Response
Reviewer #3:
- After looking at updated Table 1 it becomes prominent, that fusion landscape is quite different in adenocarcinoma and squamous cell carcinoma samples, potentially reflecting differences in biology in between these non-small cell lung cancer subtypes. Additionally, samples with higher stage and grade demonstrate higher relative percentage of fusion genes involving oncogenes and tumor suppressor genes. Authors should address these points in results and discussion sections.
Response: Thank you for your suggestions. We have added the following to the Results: “Patients with higher grade have higher relative percentage of fusion genes involving oncogenes and TSGs, but no correlation with tumor stage.” and have added the following to the Discussion: “A correlation was observed for the grade and fusion genes involving TSGs and oncogenes, but no correlation with tumor stage. Fusion gene involving TSGs and oncogenes in the tumors at higher grades were significantly more frequently than those in the tumors at lower grades. This results have also been found in prostate and brain tumors in the specific fusion genes TMPRSS2-ERG and ARHGEF2-NTRK1, which involved ERG and NTRK1 oncogenes [25,26].”
- Figure 2 occupies two pages. It seems to be beneficial to convert it into a table and move seq chromatograms to supplemental figures.
Response: Figure 2 has been removed to Figure S1, and the results have been added to Table S1.
Round 2
Reviewer 3 Report
Additional suggestions are addressed
This manuscript is a resubmission of an earlier submission. The following is a list of the peer review reports and author responses from that submission.
Round 1
Reviewer 1 Report
The authors present a retrospective study on fusion genes in non-small cell lumg carcinomas. The archival samples are used, so authors could use clinical data on prgnostic and predictive role of detected fusion genes and increased the clinical output of the study. Also treatment of the patients should be correlated with results. The specimens number is slightly limited nevetheless. Furthermore correlation with tumor stage and grade may be provided. Conclusions are too general, should show more studied data and their output.
Author Response
The authors present a retrospective study on fusion genes in non-small cell lung carcinomas. The archival samples are used, so authors could use clinical data on prognostic and predictive role of detected fusion genes and increased the clinical output of the study. Also treatment of the patients should be correlated with results. The specimens number is slightly limited nevetheless.
Response: We have added five cases to increase the number of cases.
Furthermore, correlation with tumor stage and grade may be provided.
Response: Thank you for your suggestion. We have added information on correlations with tumor stage and grade in the Results as follows: “We found that age and grade were correlated with the presence of fusion genes involving TSGs (p=0.0208 and p=0.0363, respectively), whereas grade was correlated with the presence of fusion genes involving oncogenes (p=0.0086) (Table 1).”
Conclusions are too general, should show more studied data and their output.
Response: We have added more data to the Conclusions as follows “In this study, we identified known and novel fusion genes present in NSCLC patients using RNA-Seq and integrated gene expression, mutation and fusion annotation data. Expression levels increased with ASXL1 (tumor suppressor gene), CACNA1A, EEF1A1, and RET (three oncogenes) fusions. The tumor grade and age were associated with fusion genes containing TSGs and tumor grade was also associated with fusion genes containing oncogenes. Nine patients carried both mutations and fusions of CDGs. Mutual exclusivity between the two types of genomic alteration, fusions and mutations, was observed. Furthermore, our study revealed an association between EGFR mutations and the number of fusion genes in the same patient. Some cancer patients may benefit from existing drugs that target fusion protein partners, expanding the therapeutic options available. Routine searches for fusion genes in cancer patients may improve their survival by revealing additional treatment targets for precise personalized medicine.”
Reviewer 2 Report
This study examined a relatively small cohort of NSCLC patients using RNA sequencing, whole-exome sequencing, and gene expression analysis, and they identified known and novel fusion genes. The authors should consider the following:
- They should validate their RNA sequencing/WES findings with FISH and RT-PCR.
- As only 8 matched normal tissues were investigated in 49 NSCLC patients, it is judged that the investigation of gene mutation will be insufficient.
- The authors should indicate the frequency of EGFR mutation and ALK fusion gene in 49 patients with NSCLC, and investigate statistically the correlation of fusion genes and these gene alterations.
- I don't really understand what you mean. I'm glad if you mention what you mean in discussion.
- The authors should discuss the limitations of this study.
- The authors should deposit the RNA seq/WES/gene expression data into GEO database.
Author Response
- They should validate their RNA sequencing/WES findings with FISH and RT-PCR.
Response: Thank you for your suggestion. We validated 77 novel fusion genes via RT-PCR and Sanger sequencing and have updated the revised manuscript accordingly (Figure 2).
- As only 8 matched normal tissues were investigated in 49 NSCLC patients, it is judged that the investigation of gene mutation will be insufficient.
Response: Although we have only eight matched normal tissues, we identified the gene mutations using several databases (e.g., TCGA, COSMIC, and gnomAD) and analyzing tools (CADD, SIFT, and PolyPhen), as well as compared the sequenced reads to distinguish between germline and somatic variants. The results have been published in Respiratory Research 22:3, 2021.
- The authors should indicate the frequency of EGFR mutation and ALK fusion gene in 49 patients with NSCLC, and investigate statistically the correlation of fusion genes and these gene alterations.
Response: Of 54 patients, we identified 27 (50%) who harbored EGFR mutations, but no ALK fusion was found. A future study that evaluates more samples will be important for correlating ALK fusion with the EGFR alterations. We investigated the correlations between other fusion genes and EGFR alterations and have added the following sentences to the Results: “EGFR mutations play important roles in the development and treatment of lung cancer, and they occur frequently in Taiwanese people and other Asians. The relationship between EGFR mutations and fusion genes has not been explored previously. We investigated whether fusion genes are correlated with EGFR mutations. The results showed that the EGFR mutations are not associated with driver fusions (Table 3). However, we found that EGFR mutations are associated with the number of fusion genes (p=0.0038), and patients with EGFR mutations may have more fusion genes (≥ 5) than do patients with wild-type EGFR.”
Moreover, we have added the following to the Discussion: “We also explored the correlation between EGFR mutations and the number of fusion genes in each case. The results showed that cases with EGFR mutations had more fusion genes. The association between the occurrence of fusion genes and the EGFR genetic background may result from tumors with an EGFR mutation exhibiting increased cell proliferation and survival, which then increases the likelihood of fusion genes occurring [25].”
- I don't really understand what you mean. I'm glad if you mention what you mean in discussion.
Response: We have clarified our meaning in the Discussion as follows: “Our study explored the clinical significance of fusion genes in lung cancer for the first time in Taiwan. We examined whether the concepts of driver and passenger fusions are useful in explaining the importance of fusion genes in the development of lung cancer, just as driver and passenger mutations. Therefore, we classified novel fusions as driver or passenger fusions according to the number of detected cancer driver gene variants. If a patient has fewer than the average number of gene variants and no known driver variants, any fusion gene observed is called a driver fusion. By contrast, if the patient has more than the average number of gene variants, especially known driver variants, any fusion gene observed is called a passenger fusion.”
- The authors should discuss the limitations of this study.
Response: We have added limitations of the study to the Discussion as follows: “This study has the following limitations. First, the cohort size was too small to make more definitive conclusions and obtain statistical significance in the correlation analysis; several gene fusions were found in three or fewer patients, making population-wise interpretation almost impossible. Second, we did not perform functional studies to confirm the effects of driver fusion genes, which might have influenced our subgrouping accuracy. Third, the study is based solely on the Taiwanese population. The results should be re-evaluated in other studies based on larger heterogeneous cohorts, to further validate our findings.”
- The authors should deposit the RNA seq/WES/gene expression data into GEO database.
Response: The RNA-Seq and WES data from this study were submitted to the NCBI Sequence Read Archive (SRA) under BioProject accession nos. PRJNA698419 and PRJNA698277, respectively.
Reviewer 3 Report
Reviewer suggestion – reject.
Overall, this is a well-written manuscript, describing work of Chang et al who applied RNA-seq analysis to investigate the presence of different fusion genes in a cohort of non-small cell lung cancer samples, in Taiwanese population. Authors were able to detect the presence of 16 previously reported fusions, and 17 novel ones in their cohort. The study has a certain novelty degree, however, there are numerous limitations that decrease the enthusiasm.
Major issues:
- Cohort size is too small (which is not discussed by the authors) to obtain any definitive conclusion and get a statistical significance in correlation analysis, several genes fusions are obtained only in 1 or two patients, which makes population-wise interpretation almost impossible. Additional limitation is that the study is based on solely Taiwanese population (which should be reflected in the title), and should be carefully compared to studies, that are based on heterogeneous/larger cohorts. Again, authors do not address this in discussion.
- Analysis of clinical significance did not demonstrate any statistically significant correlation results between characterized fusions and clinical outcomes. Age and stage correlations are borderline, and given the number of samples, are not convincing, so the clinical relevance of fusions detected remains obscure.
- Analysis of mutual exclusivity of gene fusions and driver gene mutations is very concerning. Authors bulked up together 299 known cancer driver genes and checked whether these are simultaneously present or not with fusion genes of interest in the samples. This is a very general approach, which needs to be reworked. It is not correct, given the complexity of tumor biology and oncogenesis in general, which is characterized as step-by-step alteration in oncosupressors and accumulation of driver mutations, to nominate every single gene out of these 299 candidates, as a potential oncodriver and compare if a damaging mutation in this gene is present along with fusion alteration or not.
- Conclusion suggesting NOTCH2 fusion implicating survival in a patient with EGFR mutant NSCLC is overestimated. This is just one case.
- Study lacks any substantial benchmarking analysis to a larger cohort using publicly available RNA-seq repositories data (e.g. TCGA), which is crucial given the limitations of the study and lack of any definite conclusions.
Other issues:
- Refs 1, 7 are outdated and newer studies are available. Ref 8 has no authors listed.
- Last sentence of the second paragraph (lines 61-62) is not correct and needs to be rephrased. The problem is not the lack of evidence for clinical effectiveness, but emergence of secondary resistance to targeted therapies.
- Table illustrating clinical characteristics of the cohort should be expanded. Authors should update diagnosis according to up-to-date WHO classification, add TNM staging, info on how the samples were obtained (e.g. surgery, bronchoscopy etc.), treatment info.
- First paragraph of the discussion section is yet another summary of the results and can be omitted.
Author Response
Major issues:
- Cohort size is too small (which is not discussed by the authors) to obtain any definitive conclusion and get a statistical significance in correlation analysis, several genes fusions are obtained only in 1 or two patients, which makes population-wise interpretation almost impossible. Additional limitation is that the study is based on solely Taiwanese population (which should be reflected in the title), and should be carefully compared to studies, that are based on heterogeneous/larger cohorts. Again, authors do not address this in discussion.
Response: Thank you for your comment. We have added five cases to increase the number of cases. We have also added study limitations to the Discussion as follows: “This study has the following limitations. First, the cohort size was too small to make more definitive conclusions and obtain statistical significance in the correlation analysis; several gene fusions were found in three or fewer patients, making population-wise interpretation almost impossible. Second, we did not perform functional studies to confirm the effects of driver fusion genes, which might have influenced our subgrouping accuracy. Third, the study is based solely on the Taiwanese population. The results should be re-evaluated in other studies based on larger heterogeneous cohorts, to further validate our findings.”
The title has been changed as suggested and now reads as: “The fusion gene landscape in Taiwanese patients with non-small cell lung cancer”.
- Analysis of clinical significance did not demonstrate any statistically significant correlation results between characterized fusions and clinical outcomes. Age and stage correlations are borderline, and given the number of samples, are not convincing, so the clinical relevance of fusions detected remains obscure.
Response: After we added five cases, we were able to find fusion genes involving either TSG or an oncogene that are significantly correlated with high tumor grades (p=0.0363 and p= 0.0086, respectively), as well as fusion genes involving TSGs that are correlated with age (p=0.0208). These results showed that the correlations were borderline compared with the original analysis. We think that some fusion genes may have clinical significance, and functional studies are needed to confirm their impacts, as in the study by Lu et al. [22]. We have added this as a limitation in the Discussion.
- Analysis of mutual exclusivity of gene fusions and driver gene mutations is very concerning. Authors bulked up together 299 known cancer driver genes and checked whether these are simultaneously present or not with fusion genes of interest in the samples. This is a very general approach, which needs to be reworked. It is not correct, given the complexity of tumor biology and oncogenesis in general, which is characterized as step-by-step alteration in oncosupressors and accumulation of driver mutations, to nominate every single gene out of these 299 candidates, as a potential oncodriver and compare if a damaging mutation in this gene is present along with fusion alteration or not.
Response: We thank the reviewer for pointing this out. The mutual exclusivity of gene fusion and driver gene mutation usually involves the same gene and rarely involves two different genes. These results were found in Gao et al. [7], who examined 9624 tumors across 33 cancer types. We obtained similar findings with our lung cancer cases. The results of Gao et al. also suggest that many tumors are driven primarily or solely by fusion events. We agree with the reviewer’s suggestion about carcinogenesis, and further functional studies to confirm the driver fusion genes are needed. We have added this as a limitation in the Discussion.
- Conclusion suggesting NOTCH2 fusion implicating survival in a patient with EGFR mutant NSCLC is overestimated. This is just one case.
Response: We have deleted the sentence and explained this as one of the limitations.
- Study lacks any substantial benchmarking analysis to a larger cohort using publicly available RNA-seq repositories data (e.g. TCGA), which is crucial given the limitations of the study and lack of any definite conclusions.
Response: We used multiple database annotation from CTAT_HumanFusionLib (https://github.com/FusionAnnotator/CTAT_HumanFusionLib/wiki), which was included in the STAR-Fusion reference. The resources used to review our findings include ChimerDB2, ChimerKB, ChimerPub, ChimerSeq, Cosmic, YOSHIHARA TCGA [31], Klijin CellLines [32], GUO2018CR TCGA [7], TumorFusionNAR2018 [13], TCGA StarF2019 (fusions found by STAR-Fusion v1.5.0 across the corresponding TCGA datasets), and CCLE StarF2019 (fusions identified in the Cancer Cell Line Encyclopedia RNA-Seq datasets) (Tables S2 and S3).
Other issues:
- Refs 1, 7 are outdated and newer studies are available. Ref 8 has no authors listed.
Response: We have changed references 1, 7, and 8 as suggested.
- Last sentence of the second paragraph (lines 61-62) is not correct and needs to be rephrased. The problem is not the lack of evidence for clinical effectiveness, but emergence of secondary resistance to targeted therapies.
Response: The sentence has been rewritten as follows: “However, resistance to cancer therapies remains a major clinical problem and involves all models of treatment, including molecular-targeted therapy and immunotherapy.”
- Table illustrating clinical characteristics of the cohort should be expanded. Authors should update diagnosis according to up-to-date WHO classification, add TNM staging, info on how the samples were obtained (e.g. surgery, bronchoscopy etc.), treatment info.
Response: We have added grades in Table 1, as suggested. Moreover, we have added the following footnote to Table 1: “All subjects underwent surgical resection. Patients were diagnosed according to the 2015 World Health Organization and International Association for the Study of Lung Cancer guidelines. Staging was based on the guidelines of the eighth edition of the TNM classification for NSCLC.”
- First paragraph of the discussion section is yet another summary of the results and can be omitted.
Response: We have omitted first paragraph of the Discussion as suggested.